# The evolution and co-evolution of a primary care cancer research network: From academic social connection to research collaboration

Debbie Vermond[1]*, Esther de Groot[1], Valerie A. Sills[2], Georgios Lyratzopoulos[3], Fiona M. Walter[2,4], Niek J. de Wit[1], Greg Rubin[5]

1 Julius Center for Health Sciences and Primary Care, Utrecht University, University Medical Center Utrecht, Utrecht, Netherlands, 2 Department of Public Health and Primary Care, The Primary Care Unit, University of Cambridge, Cambridge, United Kingdom, 3 Department of Behavioural Science and Health, UCL, London, United Kingdom, 4 Wolfson Institute of Population Health, Barts and The London School of Medicine and Dentistry Queen Mary University of London, London, United Kingdom, 5 Population Health Sciences Institute, University of Newcastle, Newcastle, United Kingdom

* D.Vermond@umcutrecht.nl

**Data Availability Statement:** Data cannot be shared publicly to protect the privacy of our participants. In smaller scale social network

## Abstract

Academic networks are expected to enhance scientific collaboration and thereby increase research outputs. However, little is known about whether and how the initial steps of getting to know other researchers translates into effective collaborations. In this paper, we investigate the evolution and co-evolution of an academic social network and a collaborative research network (using co-authorship as a proxy measure of the latter), and simultaneously examine the effect of individual researcher characteristics (e.g. gender, seniority or workplace) on their evolving relationships. We used longitudinal data from an international network in primary care cancer research: the CanTest Collaborative (CanTest). Surveys were distributed amongst CanTest researchers to map who knows who (the 'academic social network'). Co-authorship relations were derived from Scopus (the 'collaborative network'). Stochastic actor-oriented models were employed to investigate the evolution and co-evolution of both networks. Visualizing the development of the CanTest network revealed that researchers within CanTest get to know each other quickly and also start collaborating over time (evolution of the academic social network and collaborative network respectively). Results point to a stable and solid academic social network that is particularly encouraging towards more junior researchers; yet differing for male and female researchers (the effect of individual researcher characteristics). Moreover, although the academic social network and the research collaborations do not grow at the same pace, the benefit of creating academic social relationships to stimulate effective research collaboration is clearly demonstrated (co-evolution of both networks).

## Introduction

Until recently, efforts to improve diagnostic accuracy for cancer were based on enlarging capacity in secondary care. This can lead to longer access times, higher costs and greater

analysis studies, who are the respondents, even when using anonymized data, may be obvious to potential readers. Data are available from the data management department of the Julius Center, University Medical Center Utrecht (dm_julius@umcutrecht.nl) for researchers who meet the criteria for access to confidential data.

**Funding:** This research arises from the CanTest Collaborative, which is funded by Cancer Research UK (https://www.cancerresearchuk.org/) [C8640/A23385], of which DV is a PhD student, EdG is a Postdoctoral Researcher, FW is Director (received the award), GR, GL & NdW are Associate Directors and VS is Programme Manager. The funder had no role in study design, data collection and analysis, decision to publish, or preparation of the manuscript.

**Competing interests:** The authors have declared that no competing interests exist.

risks of error and delay [1,2]. Increasingly, primary care is regarded as the optimal setting to initiate health care improvements [3,4]. Timely and adequate diagnosis in primary care is vital for improving diagnostic accuracy in cancer, and therefore more research capacity focused on diagnostic testing in the primary care setting is required [5]. Scientific progress may particularly benefit from multi-disciplinary collaborations between researchers, across different research institutes and countries, as well as across the entire continuum from test development to clinical implementation. Indeed, collaboration between researchers is known to increase scientific productivity and the quality of research compared to individual research efforts [6–9].

Across academic disciplines, networks are developing to connect researchers worldwide and underpin scientific progress [10–12]. In these networks, researchers establish relationships through a variety of social-academic activities and platforms. The CanTest Collaborative (CanTest) is a clear example of such a network and serves as case study for this manuscript (https://cantest.org). CanTest was formally constituted in 2017 with funding from Cancer Research UK, building upon several individual collaborations between senior primary care cancer researchers in its participating centres. It comprises nice academic centres in five different countries and across three continents; individual researchers from 10 other academic centres are also involved by invitation [13]. Its main objectives are to increase capacity and sustainability of research into early detection and diagnosis of cancer—recruiting and supporting the development of a new generation of researchers to establish themselves—and to assess and evaluate approaches to improving early detection and diagnosis of cancer in primary care (the work carried out in this study will shed insight in how CanTest has addressed the first objective). By spanning disciplinary, organisational and national boundaries, academic social networks (networks of researchers connected by informal interactions and social relationships) such as CanTest can capture the social substrate of scientific productivity and promote interactions among researchers that facilitate the sharing of meaning and completion of their tasks [14]. Ultimately, through increased social support and better access to critical resources, membership and active participation in such networks is considered to enhance collaboration and scientific outputs (such as joint projects and co-authorship), as well as the individual development of more junior researchers [9,15,16].

However, academic social networks do not necessarily translate into effective collaborative research networks (networks of researchers connected by collaboration); individual characteristics of researchers in networks are considered to play a significant role in their development [17–19]. Two characteristics of researchers that have been thoroughly studied are gender and seniority [20,21]. Females are more often reported to appreciate relationships and the process of collaboration, whereas actual collaborative activity is reported to be higher for males [22–24]. Comparably, where junior researchers are more likely to increase their number of academic social relationships to gain access to new resources for collaboration, senior researchers may be more reserved in creating new relationships because they have many collaborative relationships already [25–27]. Yet, studies that link academic social relationships and collaboration—to explore the effect of gender and seniority in academic social relationships and collaboration in more detail—are lacking.

Another factor that may play a prominent role in explaining the development of networks is individual network positions [28,29]. Along with the individual characteristics of researchers, their positions in relation to each other may steer their relationships. In research on academic collaboration, the logic of network embeddedness (friends of friends tending to become friends) and preferential attachment (individuals seeking out relationships preferentially with others who are popular already) has demonstrated that researchers tend to connect with the connections-of-their-connections and with well-connected researchers [30–33]. Yet, again,

while existing studies looking at the impact of network positions on collaboration have provided a variety of important insights, we still have limited understanding of how individual positions and individual characteristics relate to each other while conditioning academic relationships and collaboration.

Although there is increased financial support for academic networking, there is little published evidence that this type of research collaboration actually accelerates research output [34,35]. Previous research studies have explored separately the development of academic relationships and collaboration (the separate 'evolution' of two networks). Yet, although academic interaction is considered an important factor in collaboration, we still have limited understanding of how the expansion of academic social networks translates into effective research collaboration, i.e. to what extent one leads to the other (the 'co-evolution' of two networks) [36,37]. Fig 1 visualizes both processes, with the evolution aspect depicted on the vertical axis (solid box) and co-evolution on the horizontal axis (outline box). Evolution is about how a network changes over time, whereas co-evolution describes how changes in one network impact the other network (i.e. how academic social relationships influences collaboration). The objectives of this study are to increase knowledge on (1) the evolution of academic social networks and collaborative networks, (2) the co-evolution of academic social networks and collaborative networks, and (3) the influence of individual researcher characteristics and their network positions on evolution and co-evolution.

## Methods

### Context

To assess the transition of academic social networks into effective research collaborations, we collected longitudinal network data from an ongoing, international network in primary care cancer research—CanTest. This international research collaborative facilitates international collaboration in primary care cancer research through (1) promoting joint research, (2)

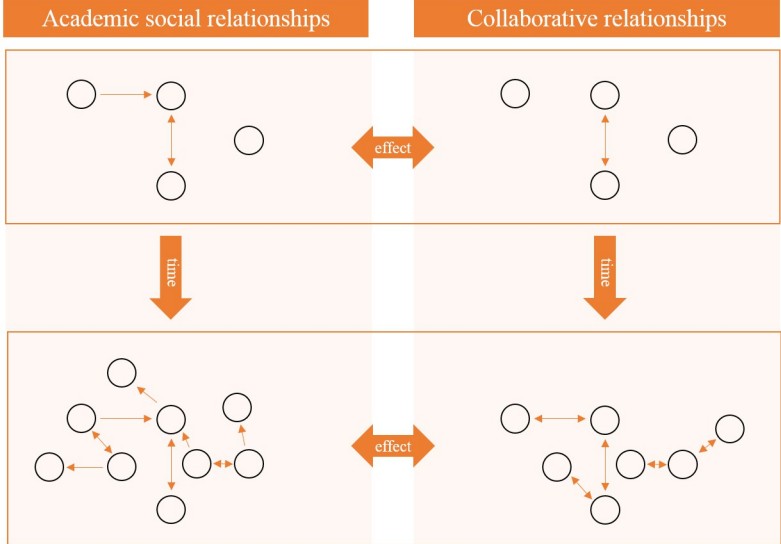

**Fig 1. Evolution and co-evolution of the academic social network and collaborative network.** The vertical axis shows the evolution of the academic social network and the collaborative network, with two different snapshots in time (top and bottom) for the purpose of illustration. The horizontal axis shows the co-evolution of both networks.

providing various training opportunities and (3) boosting academic exchange. CanTest organizes numerous networking events, the most notable of which is the annual CanTest International School. During this week-long residential activity, researchers in the network are brought together to exchange knowledge and experiences. Two of these International Schools were held in 2018 and 2019. Due to the impact of the coronavirus epidemic, virtual networking events took the place of the 2020 School, and a virtual School was held in 2021. While some network interactions are formally organized, other more spontaneous forms of interaction are also encouraged [38].

## Data

Data on the academic social relationships between all 60 researchers in the CanTest network at the time of this study were collected through online surveys at three points in time:

- T0—Point of entry to CanTest (April 2017/2018)—before the first International School

- T1—April 2019—just before the second International School

- T2—June 2019—just after the second International School.

Informed consent was obtained at the start of each survey (S1 Appendix). As is customary in network research, one matrix question, or roster, explored the relationships between researchers in the CanTest network ('academic family') at these three points in time: "*Which other members of the CanTest family do you know professionally, and how did/do you connect and interact with them*?" [39]. From this roster of all researchers in the network, researchers were asked to identify other researchers with whom they were connected. For each of the 60 researchers in the network, they selected either 'yes' or 'no' for any of four possible ways of being connected: (1) exchanged emails or WhatsApp messages, (2) met at a conference, workshop, meeting or training event, (3) involved in the same individual project, and (4) working at the same institution or department (S1 Appendix). For the analysis, it was necessary to aggregate the results; we did so by recoding any number of ways of being connected (either by 1, 2, 3 or 4 items) as 1 = "yes" and recoding the absence of any connection as 0 = "no". The network was directed so a distinction was made between outgoing relationships ($i$ knows $j$) and incoming relationships ($j$ knows $i$). Moreover, since "knowing someone" is assumed to be irreversible, existing relationships could not be terminated but were necessarily maintained. Prior to sending the network survey to the CanTest researchers, the face validity of the survey was assessed by a group of native speakers. They reviewed the survey for ease of use and understanding. All of them deemed the network survey to be acceptable.

We used co-authorship as a proxy measure for research collaboration. Data on co-authorship relations between researchers in the network were derived from Scopus. The time windows searched for the three points in time were 'up to entry to CanTest' (T0), 2018 (T1) and 2019 (T2). Scopus was searched by Author Identifiers and relevant key words (TITLE-ABS-KEY(cancer* OR tumour* OR tumor* OR neoplasm* OR malignan* OR carcinoma* OR sarcoma* OR melanoma* OR lesion* OR leukaemia OR leukemia OR lymphoma* OR myeloma*)) to collect data on the co-authorship relationships between the researchers in the network. Using the Author Identifiers, we corrected for different spelling of researcher names, and merged them when one researcher turned out to have different Author Identifiers. Thereafter, data were entered in a matrix of size 60x60, each row and column representing a researcher. A co-authorship relation between two researchers was coded as 1 = "yes", and absence of a co-authorship relation was coded as 0 = "no". The co-authorship network

## Box 1. Definitions

| | |
|---|---|
| Academic social network | Network of researchers connected by informal interactions and social relationships |
| | *Who do you know?* — self-reported |
| Collaborative network | Network of researchers connected by co-authorship as a proxy for collaboration |
| | *Who do you collaborate with?* — database-derived |

was non-directed (connections between co-authoring researchers are by definition reciprocal) so there is no distinction between incoming and outgoing relationships. Box 1 summarizes how the definitions of the academic social network and the collaborative network were operationalised.

The CanTest member register was consulted to collect individual researcher characteristics. Gender was treated as a constant, categorical actor covariate and was coded as either 0 = "female" or 1 = "male". Data on researcher seniority was treated as a changing, categorical actor covariate and was coded as 0 = "junior researcher" (early stages of PhD or pre-PhD), 1 = "early-career researcher" (later stages of PhD or early post-doc; also pre-PhD with multiple first author publications in the cancer domain), 2 = "mid-career researcher" (more experienced post-doc; three or more first/last author publications in the cancer domain, supervising more junior researchers/been awarded personal grant(s)), or 3 = "senior researcher" (senior lecturer and above; e.g. been awarded an institutional grant, managing a research group, senior lecturer status). In addition, data on physical workplace (i.e. country and institute) and professional background (i.e. researcher or clinical researcher were extracted and added to the model to control for sources of scientific embeddedness [8,35,40–44]. Data on the physical workplace covered 18 research institutes across five countries (UK, Denmark, USA, Australia and the Netherlands). The professional background of each researcher was coded as either 0 = "researcher" or 1 = "clinical researcher". All data were kept in a locked file cabinet and were anonymized prior to analysis to maximize confidentiality.

### Model for analysis

We investigated the process of network evolution and co-evolution using stochastic actor-oriented models [45–47]. Statistical analysis of longitudinal network data is not possible with conventional statistical methods assuming independence of observations because, in networks, changing connections are typically interrelated with other simultaneous processes (i.e. changes in other connections in the same network or characteristics of the individual researchers involved). Stochastic actor-based models use a combination of simulation methods with statistical model fitting. For this study, models were estimated with the data-analysis package SIENA in R (Simulation Investigation for Empirical Network Analysis), which is suitable for binary social network data in which a pair of researchers is represented in either state 1 (relationship) or 0 (no relationship) [48]. For all models, t-ratios (indicators of convergence) were obtained of less than 0.1, and overall convergence of less than 0.25, which signals good model convergence [48]. Goodness of fit was assessed with auxiliary statistics (outdegree distribution, indegree distribution and triad census) and was deemed acceptable [49].

To explore the evolution and co-evolution of both networks, we created three models. Model 1 and 2 capture the separate evolution of the academic social network and the collaborative network respectively. Model 3 captures the co-evolution of both networks, exploring the influence of the academic social network on the collaborative network (to measure how both networks are inter-related, considering that (1) some initiatives, although intended to become publications, do not progress, and (2) there are lag phases between informal interaction and publication). All three models contain a combination of effects to control for both individual researchers' positions and an individual researcher's characteristics. Effects express, for example, whether researchers are likely to get to know the connections of their connections (network embeddedness; transitivity), or whether researchers with many relationships are more likely to have additional relationships over time (preferential attachment; indegree popularity). Fig 2 visualizes the effects included in the three models. Detailed explanations of these effects and whether these effects are present in each of the three models are provided in S2 Appendix.

## Results

### Network development

The development of the academic social network and the collaborative network is depicted in Fig 3, showing a rapidly growing and very dense academic social network, and a collaborative network that is less dense and grows more slowly. Additionally, Fig 4 shows a more detailed view of development of the academic social network, showing the seniority and country of each researcher. A description of the development for both networks is provided in Table 1. The average number of (outgoing) relationships for both networks increased over time, revealing researchers got to know each other as well as starting to collaborate. Furthermore, the academic social network showed a strong tendency toward reciprocity (the co-authorship network is reciprocal by definition). Clusters of researchers were present in both networks.

Next, the association between the networks is given in Table 2. It shows the correlation between the number of (outgoing) relationships for both networks, for each observation moment. These numbers can be regarded as indications of the association between the

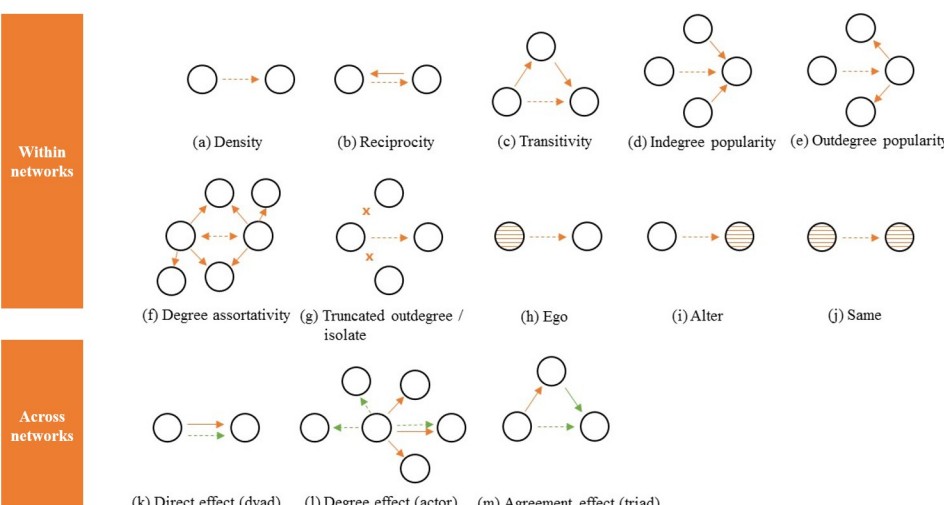

**Fig 2. Effects used in the three models.** Adapted from Stadtfeld et al [50]. The continuous arrows represent existing relationships at the start of this study; the dashed arrows represent new relationships created over the course of this study. For the cross-network effects, the difference between relationships in the two networks is represented by different colored arrows.

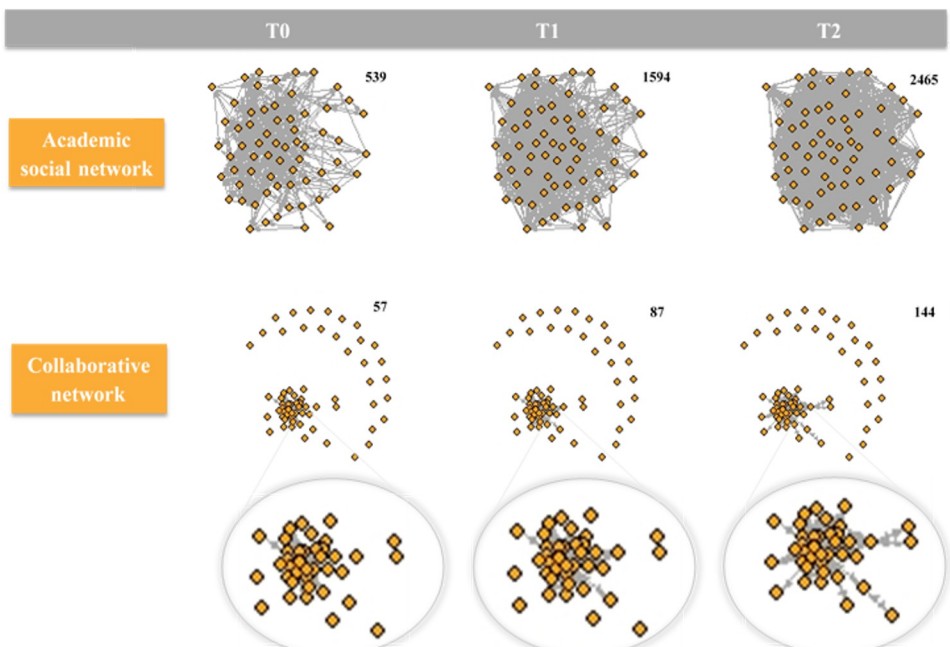

**Fig 3. Development of the academic social network and the collaborative network.** The top three figures, from left to right, visualize the development of the academic social network over time (i.e. from April 2017/2018 up to June 2019). The middle three figures, from left to right, visualize the development of the collaborative network—zooming in on the core of the network in the bottom three figures. The numbers in the top right corner of each figure represent the number of connections (total outdegree) in each figure.

development of the two networks. The correlations were positive, but decreasing over time, again reflecting a difference in the pace at which both networks developed. In addition, the association at the relationship-level (how many relations between researchers in the academic social network are also present in the collaborative network and vice versa) can be expressed by the Jaccard similarity index. This is a measure of similarity between two sets of data, formally defined as the number of connections in both networks divided by the number in either network, with higher values being indicative for higher similarity. The Jaccard similar index for each of the three observations was 0.19, 0.13, and 0.11 [47,51]. If independence between

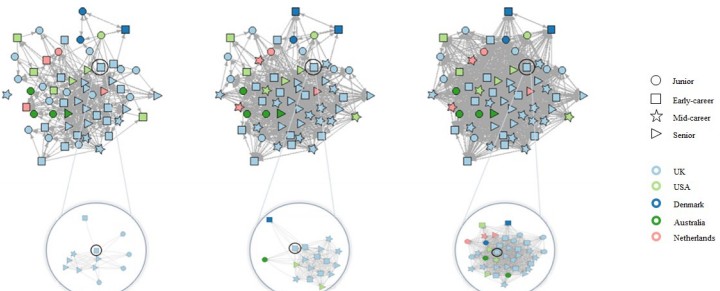

**Fig 4. Development of the academic social network showing the seniority and country of each researcher.** Again, the top three figures, from left to right, visualize the development of the academic social network over time (i.e. from April 2017/2018 up to June 2019). The relationships of one anonymized, early-career researcher from the UK have been highlighted in the bottom three figures to better illustrate how researchers of different seniority and country get to know each other over time.

**Table 1. Descriptives for the evolution of both networks.**

| | Definitions | Academic social network | | | Collaborative network | | |
|---|---|---|---|---|---|---|---|
| | | $T_0$ | $T_1$ | $T_2$ | $T_0$ | $T_1$ | $T_2$ |
| **Average (out)degree[1]** | Average number of (outgoing) relationships amongst researchers | 9.0 | 20.2 | 39.5 | 1.9 | 2.0 | 3.4 |
| **s.d. (out)degree[2]** | Standard deviation of (out)degrees | 6.8 | 11.4 | 13.1 | 3.5 | 3.8 | 5.1 |
| **Reciprocity[3]** | If i is connected to j, what is the probability that j is also connected to i? | 0.65 | 0.73 | 0.79 | - | - | - |
| **Transitivity[4] (clustering)** | If i is connected to j, and j to k, what is the probability that i is also connected to k? | 0.51 | 0.66 | 0.84 | 0.53 | 0.44 | 0.65 |

[1] An average outdegree of 9 means that researchers know on average 9 other researchers;

[2] An s.d. outdegree of 6.8 means that on average the difference between the average outdegree and individual measurements is 6.8;

[3] A reciprocity of 0.65 means that 65% of the connections between researchers are mutual;

[4] A transitivity of 0.51 indicates that 51% of the connections were transitive (referring to network embeddedness/transitivity).

the networks was assumed, the expected Jaccard coefficients would be 0.03, 0.04, and 0.08. The observed values, while not very high, were higher.

## Network evolution

The two central columns of Table 3 report the parameter estimates for the separate evolution of both networks with their associated standard errors. Irrelevant effects for the different models are coloured grey. The academic social network (Model 1) exhibited positive *reciprocity* and *transitivity* parameters, indicating that researchers over time tended to get to know (1) those researchers who they were known by, and (2) the 'friends of their friends' (i.e. they are likely to cluster in groups). Yet, the negative interaction between reciprocity and transitivity indicates that, compared to one-to-one relationships, in clusters scientists are less likely to reciprocate relationships. The positive *indegree popularity* signifies that well-known researchers are inclined to become even more well-known over time. Conversely, the negative *outdegree popularity* parameter indicates that researchers who know many others are not necessarily well-known to others as well.

With increasing seniority, researchers are less likely to get to know additional researchers within the network (negative *seniority ego*). Similarly, males are less likely to get to know others (negative *gender (male) ego*). The interaction between gender and seniority (*gender (male) x seniority (senior)*) further signifies that males of higher seniority are less likely to get to know additional others. Finally, significant positive effects for *same gender* and *same professional background* assume that researchers are more likely to get to know others of the same gender and of the same professional background.

The negative *degree (density)* parameter for the collaborative network (Model 2) indicates that researchers were selective in their collaborative relationships. Yet, having collaborative relationships increases the initiation of new collaborations (positive *in = outdeg. popularity/ activity*)—especially with low-collaborators (negative *degree assortativity*). Although male researchers were less likely to get to know others compared to females, they are more likely to

**Table 2. Correlations between the (out)degrees of the two networks, for the three time points separately.**

| | $T_0$ | $T_1$ | $T_2$ |
|---|---|---|---|
| | Collaborative network | Collaborative network | Collaborative network |
| **Academic social network** | 0.71 | 0.39 | 0.14 |

**Table 3. Evolution and co-evolution of the academic social network and the collaborative network: Parameter estimates and standard errors of SIENA models.**

| | Evolution | | | | Co-evolution | |
|---|---|---|---|---|---|---|
| | Model 1 Academic social network | | Model 2 Collaborative network[1] | | Model 3 Academic social network → Collaborative network | |
| | par. | (s.e.) | par. | (s.e.) | par. | (s.e.) |
| **Within network** | | | | | | |
| Degree (density) | | | -3.134*** | (0.375) | -5.133*** | (1.080) |
| Reciprocity | 5.324*** | (1.538) | | | | |
| Transitivity[2] | 3.200** | (1.201) | 2.232*** | (0.362) | 2.575*** | (0.669) |
| Indegree popularity[3] | 0.464* | (0.195) | | | | |
| Outdegree popularity[3] | -0.201** | (0.065) | | | | |
| In = outdeg. popularity/activity | | | 0.112*** | (0.025) | 0.152** | (0.055) |
| Degree assortativity | | | -0.311*** | (0.089) | -0.432** | (0.157) |
| Outdegree <10 | -12.063*** | (1.197) | | | | |
| Network isolate | | | -0.749 | (0.867) | -1.688 | (1.533) |
| Gender (male) ego | -3.787*** | (0.967) | | | | |
| Gender (male) alter | 0.041 | (0.079) | | | | |
| Gender (male) ego+alter | | | 0.254* | (0.119) | 0.247 | (0.167) |
| Same gender | 0.259*** | (0.081) | -0.095 | (0.139) | -0.242 | (0.192) |
| Seniority (senior) ego | -1.708*** | (0.505) | | | | |
| Seniority (senior) alter | 0.001 | (0.031) | | | | |
| Seniority (senior) alter[3] | -0.005 | (0.036) | | | | |
| Seniority (senior) ego+alter | | | -0.031 | (0.044) | -0.016 | (0.055) |
| Seniority (senior) ego+alter[3] | | | 0.077 | (0.057) | 0.086 | (0.066) |
| Seniority similarity[3] | -0.008 | (0.016) | -0.043 | (0.032) | -0.056† | (0.034) |
| Same professional background | 0.106† | (0.062) | 0.158 | (0.131) | -0.037 | (0.154) |
| Same country | -0.044 | (0.110) | 0.112 | (0.129) | 0.026 | (0.163) |
| Same institution | 2.395*** | (0.347) | 0.847*** | (0.205) | 0.529 | (0.236) |
| **Between-network: direct effect** | | | | | | |
| Academic social network | | | | | 2.554** | (0.896) |
| **Between-network: degree effects** | | | | | | |
| Outdegree activity | | | | | -0.373 | (0.278) |
| Indegree popularity | | | | | 0.296 | (0.265) |
| **Between-network: agreement** | | | | | | |
| Academic social network | | | | | -0.010 | (0.077) |
| **Interactions** | | | | | | |
| Gender (male) x seniority (senior) | -3.962*** | (1.007) | 0.151 | (0.182) | 0.028 | (0.266) |
| Transitivity x reciprocity | -2.149*** | (0.782) | | | | |

par. = parameter for the effect (estimate); (s.e.) = standard error;

† $p < 0.1$;

* $p < 0.05$;

** $p < 0.01$;

*** $p < 0.001$.

[1] For the independency assumption of the stochastic actor-oriented model, only papers with a maximum of five co-authors from within CanTest were included.

[2] gwespFF for Model 1 and gwesp for Model 2/3.

[3] Square-root transformed.

collaborate (positive *gender (male) ego+alter*). Finally, collaboration is encouraged by working in the same institution (positive *same institution*).

## Network co-evolution

The results for the co-evolution of the academic social network and collaborative network are reported in the right hand column of Table 3. The direct effect of a relationship in the academic social network on the likelihood of a relationship in the collaborative network was positive and significant (*between network*: *direct effect—academic social network*). When researcher *i* knew researcher *j*, they were likely to start collaboration (i.e. co-authorship) over time. Other cross-network effects were not significant.

## Discussion

The CanTest network is clearly successful in connecting researchers with each other. Researchers within the network establish connections promptly and effectively (objective 1: evolution of the academic social network), allowing for considerable exchange of information and ideas to s increase capacity and support sustainability of early detection and diagnosis of cancer research. Seniority and gender seem to play a major role in the development of relationships within the network (objective 3: individual researcher characteristics), not affecting who is known or not, but who gets to know others (objective 3: individual network positions). We found that more junior researchers built and expanded their academic network, but that with increasing seniority researchers were less likely to get to know others. This closely aligns with one of the key objectives of CanTest to recruit and support a new generation of researchers to establish themselves and reach early independence [13]. Moreover, compared to male researchers, female researchers seem to expand their academic social networks faster. The interaction between gender and seniority stresses even more how males of increasing seniority are less likely to expand their academic social networks over time.

Yet, in contrast with how more junior researchers expand their academic social networks, they seem to expand their collaborative research networks more slowly (objective 1 and 3: evolution of the collaborative network and the influence of individual researcher characteristics). The positive but decreasing association between the CanTest academic social network and the collaborative network over time further confirms this (objective 2: co-evolution of the academic social network and the collaborative network). The concurrent development of effective research collaboration is indeed a time consuming process—being connected informally, e.g. by being involved in the same project, may lead to collaboration on ideas and study design, application for funding, exchange visits, conduct of research, and only finally co-authorship. Despite this lengthy sequence, our results demonstrate that the transition from social connections into research collaboration does take place: creating an academic social relationship between two researchers significantly increases their chances to collaborate. The existence of the CanTest Collaborative and the structure, events and coordinated communications that go with it, has most likely contributed to the observed results. However, it is not possible to know how the networks studied here would have evolved without the existence of CanTest. The timing of the significant increase in academic social connections since the start of CanTest in April 2017 makes it likely, though, that CanTest has been instrumental in the process of creating and accelerating informal interactions and social relationships and hence collaboration.

Findings from previous studies suggesting higher co-authorship activity for males compared to females are confirmed in the current study [23]. However, we also found evidence to support findings from other studies that females are more appreciative of collaboration and so they expand their academic social networks faster [22]. Frequently assumed tendencies for

same-gender relationships are confirmed as well; however, no evidence was found for same-gender collaborations [52]. Our finding that more senior researchers tend to collaborate more often may have played a role in this. Indeed, based on a large academic bibliographic database research, Combes and Givord (2018) argued that same-gender collaborations occur more commonly at the beginning of a researcher's career and fade with seniority [53]. Furthermore, national background didn't seem to play a role in the development of individual relationships amongst researchers, although the international collaboration does strengthen the network as such. A perhaps surprising finding was that, among participants from the same institution, CanTest seems to have boosted within-own-institution collaborations, beyond boosting between-institution ones. Therefore catalysing new 'internal' network formations seems one of the means by which the network intervention has been effective. This may reflect the fact that most modern universities often encompass multiple campus sites/buildings (the boundaries between them acting as practical barriers) and different departments (the boundaries between them acting as organisational or disciplinary barriers), which mean that the potential for within-institutional collaborations cannot be taken as a given. Therefore, the CanTest network may also have boosted collaborations that—although relating to participants working at the same institution—would not have otherwise happened.

The main strength of the current study is that longitudinal rather than cross-sectional network data of a whole research network was used for understanding network evolution and co-evolution. Specifically, we were able to control for effects from researcher's positions in the network as well as their seniority and gender across three points in time. A limitation of the study was that the study period was relatively short, and co-authorship relationships may take longer to flourish. As a result, the dynamics of collaboration in CanTest may not have been fully captured. Future research will be needed to show whether 'knowing each other" translates into "collaborating with each other" even more when considered over a longer period of time. Moreover, it should be acknowledged that collaborative research activity is not limited to co-author behaviour. Other metrics for collaborative research activity could have been used, e.g. co-funding, but co-authoring activity was preferred as it is the most common metric for collaborative research activity in the literature and its data was readily accessible.

Future research should further address the evolution and co-evolution of relationships and collaboration within (cancer) research networks; in particular between less and more senior researchers as it remains unclear whether it is the bridge function that senior researchers may have towards junior researchers, the density of the CanTest network itself (densely linked networks are more efficient at diffusing information to all their members when compared to sparsely linked groups), or a combination of both that encourages researchers to move forward [25,54]. A combination of a densely linked network and the availability of one or several 'bridging researchers'—often referred to as brokers—might be ideal, pursuing a network that is maximally effective in facilitating collaboration between its members [55]. In addition, more research into the extent to which males and females of different seniority seem to expand their networks is warranted. There is evidence for a 'saturation point' for social connections, arguing that researchers are likely to refrain from initiating new connections if they already have many connections, or for females being more likely to create more diverse social capital [53,56]. Yet, there may be countless other mechanisms that address the interplay between seniority and gender in (academic) social network formation.

Simply establishing the infrastructure for a network of researchers to get to know each other will not necessarily make them collaborate. It may be just a matter of time, but the current study—unique in using longitudinal data to study the co-evolution of social connections and collaboration considering both researcher's characteristics and positions in networks— shows how the seniority and gender of researchers are particularly worth paying attention to

when establishing effective research networks. Increased understanding of how to address and balance researcher's characteristics might help other research initiatives or funding agencies in developing effective research networks to promote research output. This study shows how facilitating and supporting a dense research network, through formal and informal network interactions, positively affects the translation from "getting to know each other" into collaboration—time will tell whether the established social connections will lead to further collaborations in the future.

## Supporting information

**S1 Appendix. Survey: Matrix question and informed consent procedure.**
(DOCX)

**S2 Appendix. Effects used in the three models.**
(DOCX)

## Acknowledgments

We thank members of the CanTest Steering Group for early discussions on this study and the participants for completing the surveys.

## Author Contributions

**Conceptualization:** Debbie Vermond, Esther de Groot, Valerie A. Sills, Georgios Lyratzopoulos, Fiona M. Walter, Niek J. de Wit, Greg Rubin.

**Formal analysis:** Debbie Vermond.

**Methodology:** Debbie Vermond, Esther de Groot, Valerie A. Sills, Georgios Lyratzopoulos, Fiona M. Walter, Niek J. de Wit, Greg Rubin.

**Writing – original draft:** Debbie Vermond, Esther de Groot, Valerie A. Sills.

**Writing – review & editing:** Georgios Lyratzopoulos, Fiona M. Walter, Niek J. de Wit, Greg Rubin.

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
