## [Decision Letter · Decision Letter 0]

10 Sep 2021

PONE-D-21-18996The evolution and co-evolution of a primary care cancer research network: from academic social connection to research collaborationPLOS ONE

Dear Dr. Vermond,

Thank you for submitting your manuscript to PLOS ONE. After careful consideration, we feel that it has merit but does not fully meet PLOS ONE’s publication criteria as it currently stands. Therefore, we invite you to submit a revised version of the manuscript that addresses the points raised during the review process.

Dear Dr Vermond;

your manuscript was reviewed by 2 experts that made several suggestions that need to be addressed before further decision can be made.

mainly that: Reviewer 1 would you like that you Re-write abstract, present, with more detail the Can-Test, label the tables and present the survey applied.

Label tables and present the survey 

Reviewer 2  would you like that you explain what you mean by  social network, repeating the analyses excluding some data involving individuals in the same project. 

Please, ready carefully the comments and address them fully. Make changes in the revised version if you find it appropriate.

We look forward to receiving your revised manuscript.

Kind regards,

Cesario Bianchi

Academic Editor

PLOS ONE

Additional Editor Comments (if provided):

Dear Dr Vermond;

your manuscript was reviewed by 2 experts that made several suggestions that need to be addressed before further decision can be made.

mainly that: Reviewer 1 would you like that you Re-write abstract, present, with more detail the Can-Test, label the tables and present the survey applied.

Label tables and present the survey

Reviewer 2 would you like that you explain what you mean by social network, repeating the analyses excluding some data involving individuals in the same project.

Please, ready carefully the comments and address them fully. Make changes in the revised version if you find it appropriate.

Reviewers' comments:

Reviewer's Responses to Questions

**Comments to the Author**

1. Is the manuscript technically sound, and do the data support the conclusions?

Reviewer #1: Partly

Reviewer #2: Yes

2. Has the statistical analysis been performed appropriately and rigorously? 

Reviewer #1: Yes

Reviewer #2: I Don't Know

3. Have the authors made all data underlying the findings in their manuscript fully available?

Reviewer #1: Yes

Reviewer #2: Yes

4. Is the manuscript presented in an intelligible fashion and written in standard English?

Reviewer #1: Yes

Reviewer #2: Yes

5. Review Comments to the Author

Reviewer #1: Abstract:

-It is desirable that a brief summary of the conclusions be described in the abstract to inform the reader if the objectives of this work have been achieved.

-Authors need to standardize the format of the word “co-evolution”. In the text there are several formats: “co-evolution”, “(co-)evolution” and “coevolution”.

Introduction

Second paragraph:

-The presentation of “The Can Test Collaborative (CanTest)” is not clear. It is important to introduce the policy, vision and mission of this working group. It is important to inform the website address of the CanTest.

-The text describes that academic centers in five different countries participate in CanTest. It is important to describe the name of the institutions and countries.

-May institutions interested in participating in this working group submit an application?

-Was the definition of the term “co-evolution” described in the text obtained from any reference or it is a definition of the authors themselves? The article #35 does not mention this word and it also does not define the term “co-evolution” for the context used in this work.

-What have been the results obtained by CanTest since its creation in 2017, for example: number of projects developed, articles published, organization of events, research exchange program.

Fifth paragraph:

-Review the work objectives: “In this study, we aim to increase knowledge on (1) the evolution of academic social networks and collaborative networks and the influence of individual researcher characteristics and positions, and (2) the co-evolution of the academic social network and the collaborative network.”

-It is not clear if any other working group, similar to CanTest, participated as partnership in this research.

-The text comments that this work will evaluate individual researcher characteristics and positions. What would be these influences?

Methods:

Date:

-Attach the form template that was used for the survey. Was this form validated before application?

-Attach the model of informed consent.

-The description of the matrix question is confusing and difficult to understand. It is suggested to build a table with the questions and possible answers.

Results:

-Table 1: The definitions could be described as a label below the table. Each item in the table could be better explained in the text or in the label text.

-I suggest a brief explanation of the Jaccard coefficient and how this coefficient is interpreted. Could these data be inserted in table 1?

-What were the time intervals T0, T1 and T2 in tables 1 and 2?

-In the item "Network co-evolution" I suggest to highlight and explain in the text the most relevant points of the table 3.

-Table 3: Write a label for the words “par.” e “(s.e.)”.

Discussion and conclusions

-A point that needs to be discussed is if the CanTest is achieving the desired goals within the context of the policy, vision and mission of this working group.

-If the goals are being achieved, what are the strengths and opportunities for improvement.

-If the objectives are not being achieved, what are the failures, weaknesses and action plans for continuous improvement of this working group.

-I suggest to review the conclusions. The text content is not in line with the objectives described in the introduction.

Reviewer #2: Thank you for the opportunity review this article. I really enjoyed reading it, it is very well written and I think it is worth publishing.

I think that the authors can clarify a few points.

1. They need to define in very clear terms what they mean by social network (which they later call academic social networks and I think they should use one term all the way through) and a collaborative networks very early on in the publication

2. More information about distinguishing between the two is required both as concepts and also in their approach to analysis (see point 3 below)

3. The data they use for the social network analysis includes if study participants are ‘involved in the same individual project’. This suggests some tautology as this same project can in fact be something that generates a publication – in fact it would be what one might expect in an academic / research environment. The two are interrelated so how did they distinguish between the two? Further as the data for the SNA are then reduced into a yes/no option this may have a very large influence on the findings.

4. It may be that my point above is dealt with by the method of analysis that they use ‘ stochastic actor-based models’ – but I doubt it. Someone who is more knowledgeable with SNA would need to comment on this. However, I can’t see how they deal with the overlap of ‘working on the same project’ with ‘publishing’ especially as working in the same project is it is combined with the other markers of the academic social network. I am wondering if repeating the analysis excluding data related to ‘involved in the same individual project’ would produce a different result?

In the discussion the authors may want to consider the following points

1. They claim that the social networking is important and ‘This closely aligns with one of the key purposes of CanTest, namely to support the next generation researchers to establish themselves and reach early independence’. However this is only the case if there is a very close correlation between the social network and publication output and I am not sure this case is made at the point at which they come to this conclusion. It seems they base this conclusion the results of social networking. I am not suggesting that they can’t come to this conclusion but it seems to me it must be related to the co-evolution argument.

2. The authors note that ‘creating an academic social relationship between two researchers significantly increases their chances to collaborate.’ This seems self-evident – it is hard to work with someone without knowing them. I suspect the point being made is that the processes that Can Test set up is instrumental in this. However without a counterfactual – other ways of people meeting (through co-incidental conferences perhaps) may be just as effective. Perhaps this could be tested if the Can Test training events (excluding conferences) was investigated.

3. I suspect what I am wondering is if reducing their data to a yes no in the SNA instead of testing each of the four individual markets of being active in a social network should be investigated?

6. PLOS authors have the option to publish the peer review history of their article (what does this mean?). If published, this will include your full peer review and any attached files.

Reviewer #1: No

Reviewer #2: No

---

## [Author Response · Author response to Decision Letter 0]

25 Mar 2022

Our extended response to the reviewer/editor comments are included in the file 'Response to reviewers'.

---

## [Decision Letter · Decision Letter 1]

18 Jul 2022

The evolution and co-evolution of a primary care cancer research network: from academic social connection to research collaboration

PONE-D-21-18996R1

Dear Dr. Vermond,

We’re pleased to inform you that your manuscript has been judged scientifically suitable for publication and will be formally accepted for publication once it meets all outstanding technical requirements.

Kind regards,

George Vousden

Staff Editor

PLOS ONE

Additional Editor Comments (optional):

Reviewers' comments:

Reviewer's Responses to Questions

**Comments to the Author**

1. If the authors have adequately addressed your comments raised in a previous round of review and you feel that this manuscript is now acceptable for publication, you may indicate that here to bypass the “Comments to the Author” section, enter your conflict of interest statement in the “Confidential to Editor” section, and submit your "Accept" recommendation.

Reviewer #1: All comments have been addressed

2. Is the manuscript technically sound, and do the data support the conclusions?

Reviewer #1: Yes

3. Has the statistical analysis been performed appropriately and rigorously? 

Reviewer #1: Yes

4. Have the authors made all data underlying the findings in their manuscript fully available?

Reviewer #1: Yes

5. Is the manuscript presented in an intelligible fashion and written in standard English?

Reviewer #1: Yes

6. Review Comments to the Author

Reviewer #1: All suggestions and proposed corrections were accepted by the authors. I have no further comments to add nor new corrections to suggest for this article.

7. PLOS authors have the option to publish the peer review history of their article (what does this mean?). If published, this will include your full peer review and any attached files.

Reviewer #1: No

---

## [Editor Report · Acceptance letter]

21 Jul 2022

PONE-D-21-18996R1 

The evolution and co-evolution of a primary care cancer research network: from academic social connection to research collaboration 

Dear Dr. Vermond:

I'm pleased to inform you that your manuscript has been deemed suitable for publication in PLOS ONE. Congratulations! Your manuscript is now with our production department. 

Kind regards, 

on behalf of

Dr. George Vousden 

Staff Editor

PLOS ONE